# TRPV4 Channels Promote Pathological, but Not Physiological, Cardiac Remodeling through the Activation of Calcineurin/NFAT and TRPC6

**DOI:** 10.3390/ijms25031541

**Published:** 2024-01-26

**Authors:** Laia Yáñez-Bisbe, Mar Moya, Antonio Rodríguez-Sinovas, Marisol Ruiz-Meana, Javier Inserte, Marta Tajes, Montserrat Batlle, Eduard Guasch, Aleksandra Mas-Stachurska, Elisabet Miró, Nuria Rivas, Ignacio Ferreira González, Anna Garcia-Elias, Begoña Benito

**Affiliations:** 1Cardiovascular Diseases Research Group, Vall d’Hebron Institut de Recerca (VHIR), Hospital Universitari Vall d’Hebron, 08035 Barcelona, Spain; laia.yanez@vhir.org (L.Y.-B.); antonio.rodriguez.sinovas@vhir.org (A.R.-S.); marisol.ruizmeana@vhir.org (M.R.-M.); javier.inserte@vhir.org (J.I.); elisabet.miro@vhir.org (E.M.); ignacio.ferreira@vhir.org (I.F.G.); 2Centro de Investigación Biomédica en Red de Enfermedades Cardiovasculares (CIBERCV), Instituto de Salud Carlos III, 28029 Madrid, Spain; 3Bio-Heart Cardiovascular Diseases Research Group, Bellvitge Biomedical Research Institute (IDIBELL), L’Hospitalet de Llobregat, 08908 Barcelona, Spain; mtajes@idibell.cat; 4Institute for Biomedical Research August Pi i Sunyer (IDIBAPS), 08036 Barcelona, Spain; mbatlle@clinic.cat (M.B.); eguasch@clinic.cat (E.G.); amasstachurska@psmar.cat (A.M.-S.); 5Cardiology Department, Hospital Clínic, 08036 Barcelona, Spain; 6Cardiology Department, Hospital del Mar, 08003 Barcelona, Spain; 7Cardiology Department, Hospital Universitari Vall d’Hebron, 08035 Barcelona, Spain; 8Department of Medicine, Universitat Autònoma de Barcelona, 08193 Barcelona, Spain; 9Centro de Investigación Biomédica en Red en Epidemiología y Salud Pública (CIBERESP), Instituto de Salud Carlos III, 28029 Madrid, Spain; 10Department of Clinical Research, ASCIRES-CETIR Biomedic Group, 08029 Barcelona, Spain; anna80@gmail.com

**Keywords:** TRPV4, TRPC6, calcium, mechanotransduction, mechanoreceptors, TRP, pathological remodeling, heart failure, physiological remodeling, exercise

## Abstract

TRPV4 channels, which respond to mechanical activation by permeating Ca^2+^ into the cell, may play a pivotal role in cardiac remodeling during cardiac overload. Our study aimed to investigate TRPV4 involvement in pathological and physiological remodeling through Ca^2+^-dependent signaling. TRPV4 expression was assessed in heart failure (HF) models, induced by isoproterenol infusion or transverse aortic constriction, and in exercise-induced adaptive remodeling models. The impact of genetic TRPV4 inhibition on HF was studied by echocardiography, histology, gene and protein analysis, arrhythmia inducibility, Ca^2+^ dynamics, calcineurin (CN) activity, and NFAT nuclear translocation. TRPV4 expression exclusively increased in HF models, strongly correlating with fibrosis. Isoproterenol-administered transgenic TRPV4−/− mice did not exhibit HF features. Cardiac fibroblasts (CFb) from TRPV4+/+ animals, compared to TRPV4−/−, displayed significant TRPV4 overexpression, elevated Ca^2+^ influx, and enhanced CN/NFATc3 pathway activation. TRPC6 expression paralleled that of TRPV4 in all models, with no increase in TRPV4−/− mice. In cultured CFb, the activation of TRPV4 by GSK1016790A increased TRPC6 expression, which led to enhanced CN/NFATc3 activation through synergistic action of both channels. In conclusion, TRPV4 channels contribute to pathological remodeling by promoting fibrosis and inducing TRPC6 upregulation through the activation of Ca^2+^-dependent CN/NFATc3 signaling. These results pose TRPV4 as a primary mediator of the pathological response.

## 1. Introduction

The heart is an organ with profound plasticity that ensures an adequate adaptability to changing environmental demands and stimuli. The term cardiac remodeling embraces all molecular, cellular, and interstitial changes that develop in response to conditions of chronic overload and aims to maintain an optimal cardiac performance [1]. Classically, physiological growth, a form of cardiac remodeling that is permanently balanced, such as that seen in pregnancy or exercise training, has been distinguished from adverse remodeling, which develops in response to hemodynamic stress of pressure or volume overload and, unlike the former, exhibits signs of cellular dysfunction and tissue fibrosis over time, triggering the transition to heart failure (HF) [2].

Although dichotomizing cardiac remodeling into both forms might be oversimplifying it, a number of works have suggested that some pathways seem to be distinctly activated in the physiological and the pathological response [3]. Whereas the insulin-like growth factor 1 (IGF1)-phosphoinositide 3-kinase (PI3K)-protein kinase B/Akt signaling pathway plays a pivotal role in adaptive remodeling [4], the maladaptive response is simultaneously driven by multiple pathways, majorly involving G-coupled protein receptors, the accumulation of intracellular calcium (Ca^2+^), and the activation of Ca^2+^-dependent proteins like Ca^2+^-calmodulin kinase II and calcineurin (CN), among others. This results in abnormal Ca^2+^ handling, contractile dysfunction, and fibrosis accumulation [5,6]. Despite extensive research in the field, at present, it is not well known which are the factors that determine the selective activation of these differential pathways and initiate one response or the other.

Transient receptor potential (TRP) channels form a large family of membrane cation channels involved in sensing and transmitting multiple external and internal stimuli. Several TRPs have been described in the heart, where, activated by mechanical forces, including cell stretch, allow for the entrance of Ca^2+^ (and other cations) into the cell, facilitating the activation of intracellular Ca^2+^-dependent pathways. These properties pose TRP channels as potential key mediators of the early responses that trigger cardiac remodeling, as suggested by some previous works [7,8,9]. Among them, TRPC6 channels have been shown to promote adverse remodeling both as effectors and positive regulators of the CN/NFAT signaling pathway in cardiomyocytes (CM) [8]. More recently, TRPV4 channels have been involved in fibrosis formation following myocardial infarction by modulating the Rho/MRTF-A pathway in cardiac fibroblasts (FB) [9]. However, no previous investigation has assessed their differential role in pathological versus physiological remodeling, nor the potential interplay between both channels.

The TRPV4 channel is activated by mechanical forces and permeates Ca^2+^ with higher selectivity (higher PCa^2+^/PNa^+^) than other TRP channels expressed in the heart such as TRPC6, TRPC3, TRPM3, and TRPM7 [10,11,12]. Therefore, we speculated that this channel could be a determining element in the promotion of the maladaptive response. In the present work, we evaluated the differential involvement of TRPV4 channels in both adaptive and maladaptive cardiac remodeling and demonstrate that TRPV4 activation drives exclusively the pathological response mostly through the promotion of tissue fibrosis through the activation of the Ca^2+^-calmodulin/CN/NFATc3 pathway. We also demonstrate that TRPV4 channels are necessary and contribute to TRPC6 overexpression, thereby enhancing the maladaptive response. Our results position TRPV4 as a key element in the promotion of cardiac fibrosis, and a differential and primary factor in the development of pathological remodeling.

## 2. Results

### 2.1. Differential Expression of Cardiac TRPV4 Channels in Physiological and Pathological Remodeling

TRPV4 expression was evaluated in two mouse models of physiological (induced by chronic exercise, Ex group) and pathological (induced by chronic infusion of isoproterenol, HF(iso) group) cardiac remodeling. Compared to sedentary animals, moderate exercise training in mice induced echocardiographic findings consistent with the human athlete’s heart, including significant left ventricular (LV) hypertrophy (greater interventricular septal wall thickness, IVS) and mild dilation of the LV diameter (LVD), with preserved LV ejection fraction (EF; Figure 1A, Ex vs. Sed groups). Conversely, mice receiving chronic isoproterenol infusion developed LV hypertrophy and much greater LV dilation, and importantly, LV dysfunction manifested by a significant decrease in LVEF, mimicking the human HF phenotype (Figure 1A, HF(iso) vs. Sham groups). Heart weight and CM hypertrophy were increased in both models, but particularly in the HF(iso) group (Figure 1B,C). Of note, fibrosis was exclusively found in the HF model, with a mean collagen content of 11% (Figure 1D), which is comparable to that reported in other models of HF [6]. Consistently, gene expression of fibrotic markers (collagen I and α-smooth muscle actin, α-SMA) in LV samples was only increased in HF mice (Figure 1E). Altogether, these results demonstrate that our Ex and HF(iso) models show the expected characteristics of physiological and pathological remodeling, respectively.

Both types of remodeling were accompanied by specific changes in the expression of TRP channels. Chronic exercise induced mild downregulation of TRPM4 and TRPV1, whereas the HF model was associated with TRPV2 (mild) and TRPC6 (marked) overexpression, as previously described [8,13] (Appendix A). Remarkably, TRPV4 mRNA and protein expression were upregulated in mice treated with isoproterenol compared to controls, whereas no significant differences were observed between Ex and Sed animals (Figure 1F).

To further confirm these results, TRPV4 expression was also analyzed in two other models of physiological and pathological remodeling generated in rats by exercise training (Ex group) and transaortic constriction (HF(TAC) group), respectively. As expected, exercise training resulted in CM hypertrophy with no collagen deposition, whereas rats subjected to TAC displayed greater CM hypertrophy and marked tissue fibrosis (Appendix A). Again, TRPV4 gene expression was upregulated in the HF model induced by TAC, but no changes were observed following exercise training (Figure 1G).

### 2.2. Time-Course Changes in TRPV4 Expression throughout the Development of Pathological Cardiac Remodeling

As TRPV4 upregulation is an exclusive finding of pathological remodeling, we then assessed the time-course changes in TRPV4 expression throughout the development of HF and its potential association with both hallmarks of adverse cardiac remodeling: hypertrophy and fibrosis. Using exclusively the HF(iso) mouse model, we observed that IVS wall hypertrophy was present at 14 days of isoproterenol infusion, with no further increase thereafter, whereas LV chamber dilation and EF further worsened until the final timepoint set at 28 days (Figure 2A). Cardiac mass and cellular hypertrophy were already evident after 3 days of isoproterenol infusion, with mild progression over time (Figure 2B). Conversely, collagen deposition was not significantly increased until day 7 of β-adrenergic stimulation and showed a significant time-dependent increase over time (Figure 2C,D).

The expression of TRPV4 channels showed a progressive increase over time that became significant after 14 days at the protein level (Figure 2E). As shown in Figure 2F, TRPV4 expression was only mildly correlated with cell hypertrophy (Pearson’s R = 0.38) but exhibited a very strong correlation with tissue fibrosis at all times (Pearson’s R = 0.82). Our results indicate that gene and protein overexpression of TRPV4 did not precede neither hypertrophy nor fibrosis, but had a remarkable time-course correlation with the latter, pointing to a potential role of TRPV4 in fibrosis promotion within the development of pathological remodeling.

### 2.3. Effects of TRPV4 Deletion on the Development of Pathological Cardiac Remodeling

To further demonstrate the role of TRPV4 in the development of adverse cardiac remodeling, transgenic mice with null (TRPV4−/−, KO) or preserved (TRPV4+/+, WT) expression of TRPV4 were subjected to chronic isoproterenol infusion. At baseline, TRPV4−/− mice showed a ~85% reduction in TRPV4 mRNA expression and no protein expression compared to TRPV4+/+ mice (Figure 3A, light red and light blue bars). Of note, genetic deletion of TRPV4 was not compensated for by overexpression of any other main cardiac TRP channels (Appendix A). As expected, chronic infusion of isoproterenol induced TRPV4 upregulation in TRPV4+/+ animals but not in TRPV4−/− (Figure 3A, dark red and dark blue bars).

TRPV4+/+ mice treated with isoproterenol developed the anticipated features of maladaptive remodeling, including LV wall hypertrophy, LV dilation and dysfunction, cardiomyocyte hypertrophy, and tissue fibrosis (Figure 3B–E, dark red bars). Conversely, TRPV4−/− mice under isoproterenol did not exhibit significant changes in cardiac dimensions and function compared to TRPV4+/+ mice receiving the same treatment or TRPV4−/− sham (Figure 3B,C). Furthermore, TRPV4−/− of the HF(iso) group developed less CM hypertrophy (27% reduction) and remarkably reduced cardiac fibrosis (58% reduction in collagen content) compared to their counterpart TRPV4+/+ treated with chronic isoproterenol, differences that were statistically significant (Figure 3D,E, dark red versus dark blue bars).

Arrhythmia vulnerability, another significant feature accompanying pathological remodeling, was tested in Langendorff-perfused hearts at baseline and also following regional ischemia in all study groups. Little inducibility was found in all groups at baseline, but the number of total ventricular tachyarrhythmias (VTA) after regional ischemia were significantly higher in TRPV4+/+ animals of the HF(iso) group compared to TRPV4+/+ sham and TRPV4−/− receiving isoproterenol (Figure 3F). Of note, the territory under ischemia was of similar size in all experimental groups, as confirmed by the lack of staining after perfusion with Evans Blue (Appendix A).

These findings indicate that TRPV4 deletion alleviates the effects of isoproterenol infusion on cardiac hypertrophy and especially on cardiac fibrosis, thereby providing protection against the development of adverse cardiac remodeling and inducible arrhythmias.

### 2.4. TRPV4 Overexpression in Pathological Remodeling Induces Enhanced Ca^2+^ Influx in Cardiac Fibroblasts

So far, our results demonstrated that TRPV4 channels are involved in the development of adverse cardiac remodeling, possibly by playing a major role in fibrosis formation. Supporting this idea, in our HF model, TRPV4 overexpression following isoproterenol infusion was predominantly seen in cardiac FB over CM (Figure 4A). Indeed, TRPV4 channels have been previously involved in FB activation and fibrosis formation through modulation of the Rho/MRTF-A pathway [9]. As TRPV4 is a cation channel with high Ca^2+^ affinity, we hypothesized that TRPV4 overexpression could also affect intracellular Ca*^2+^* dynamics and ultimately activate Ca^2+^-dependent pathways leading to FB activation, such as the Ca^2+^-calmodulin-dependent CN/NFATc3 pathway, long involved in adverse remodeling [14] and a recognized effector of TRPV4 activation in other cell types [15].

Ca^2+^ dynamics were thus assessed through imaging fluorescence in primary cultures grown from isolated FB of the four study groups, in the presence of specific TRPV4 activators (GSK1016790A, hypotonicity), with or without pre-incubation with the TRPV4 antagonist HC067047 and thapsigargin. The addition of thapsigargin was used to inhibit the rapid reuptake of Ca^2+^ by the endoplasmic reticulum, which would otherwise have hindered the detection of TRPV4 Ca^2+^ responses. At baseline, Ca^2+^ levels were higher in FB from TRPV4 HF animals, consistent with greater TRPV4 expression in these animals (Appendix A). Moreover, FB from TRPV4+/+ isoproterenol-treated mice exhibited a robust increase in cytosolic Ca^2+^ after stimulation with GSK1016790A (100 nM), a finding that was attenuated in TRPV4+/+ sham animals and not present in TRPV4−/− animals exposed to either saline or isoproterenol (Figure 4B, top). Similar effects were found after the cells were subjected to hypotonic stress, a physiologic activator of TRPV4 (Figure 4C, top). Importantly, FB from TRPV4+/+ mice of the HF(iso) groups pre-incubated with the TRPV4 antagonist HC067047 (10 µM) showed mitigated Ca^2+^ influx in response to either GSK1016790A (Figure 4B, bottom) or hypotonic stress (Figure 4C, bottom).

These results indicate that pathological remodeling is associated with an increased expression of TRPV4 channels that are functional, leading to a greater Ca^2+^ influx into cardiac FB when challenged with TRPV4-activating stimuli.

### 2.5. TRPV4 Activation in Pathological Remodeling Mediates Fibrosis though Activation of the CN/NFATc3 Pathway

To assess whether TRPV4-mediated Ca^2+^ entrance had an effect on the Ca^2+^-calmodulin-dependent CN/NFATc3 pathway, CN expression and activity and NFAT nuclear translocation were evaluated in FB from all experimental groups. CN activity was increased in FB from TRPV4+/+ animals exposed to isoproterenol, a finding that was not observed in TRPV4−/− FB (Figure 5A). This increase affected only the enzyme’s function, as protein expression did not show significant changes among experimental groups (Figure 5B). The phosphatase activity of CN results in a conformational change in NFATc3 that promotes its nuclear translocation. In our cells, intracellular localization of NFATc3 assessed by immunostaining revealed that NFATc3 was almost exclusively located in the cytosol in both sham groups. A remarkable nuclear translocation of NFATc3 was observed in FB from TRPV4+/+ mice of the HF(iso) group (Figure 5C, dark red bars), but translocation was greatly reduced in their counterpart, TRPV4−/− mice (Figure 5C, dark blue bars). These results indicate that in our HF model, most of the NFATc3 nuclear translocation was abolished in the absence of TRPV4. Representative microphotographs of these findings are provided in Figure 5D. As shown, TRPV4+/+ HF (iso) mice exhibited a marked increase in nuclear translocation of NFATc3, which was comparable to that observed in a positive control induced by a high-calcium medium to activate CN. This response was observed neither in the TRPV4+/+ sham nor in any of the TRPV4−/− groups. Moreover, the expression levels of acta2 and col1a1 (encoding α-SMA and collagen I, respectively), known target genes of nuclear NFAT, were upregulated in TRPV4+/+ FB exposed to isoproterenol, while this increase was not observed in any of the other groups (Figure 5E).

The results, thus far, demonstrate that in FB from TRPV4+/+ mice with HF, there exists greater CN activity, which promotes the translocation of NFATc3 into the nucleus, resulting in enhanced transcription of fibrotic genes. These findings were not present in TRPV4−/− mice. To establish whether TRPV4 channels are actual activators of the CN/NFAT pathway, additional experiments were performed in primary cultured fibroblasts from TRPV4+/+ mice exposed to the TRPV4 activator GSK1016790A (100 nM). NFAT nuclear translocation was used as a surrogate of the CN/NFAT pathway activation. As shown in Figure 5F, NFAT nuclear translocation increased after specific stimulation of TRPV4 (purple bar). These effects were not observed if FB were pre-incubated with the TRPV4 inhibitor HC067047 (10 µM, gray bar). These experiments confirm that CN/NFAT are downstream effectors of TRPV4 activation in cardiac FB, a novel finding for this specific cell type.

### 2.6. The Interplay between TRPV4 and TRPC6 in the Promotion of Pathological Cardiac Remodeling

Previous publications have demonstrated that in CM, TRPC6 overexpression promotes pathological remodeling through activation of the CN/NFATc3 pathway, and in turn, TRPC6 becomes upregulated in response to CN activation, fulfilling a circuit of positive feedback [8]. We sought to explore whether this circuit was also true for cardiac FB and whether TRPV4 channels could play a role as modulators or triggering factors given their capability to activate the CN/NFAT pathway.

We first assessed the expression of TRPC6 channels in our models of physiological and pathological remodeling. Remarkably, TRPC6 expression was unchanged in mice subjected to chronic exercise (Ex group) but increased significantly in mice of the HF(iso) group (Figure 6A and Appendix A). Similar results were found in rats subjected to moderate exercise training (Ex group) and in rats undergoing TAC (HF(TAC) group) (Figure 6B). These findings insinuate a parallel behavior of TRPV4 and TRPC6 in physiological and pathological remodeling (Figure 1F,G and Figure 6A,B, respectively). Additionally, time-course changes in TRPC6 expression appeared analogous to those observed in TRPV4 throughout the development of HF, with an increase in protein expression that became significant at 14 days (Figure 6C versus Figure 2D). As for TRPV4, TRPC6 overexpression associated with pathological remodeling was majorly present in cardiac FB over CM (Appendix A).

TRPC6 expression was also assessed in transgenic TRPV4 KO mice subjected to chronic infusion of isoproterenol (HF(iso) group) or shams. Strikingly, isoproterenol treatment induced TRPC6 upregulation in TRPV4+/+ mice but failed to do so in TRPV4−/− mice (Figure 6D), despite the fact that genetic deletion of TRPV4 did not have effects on TRPC6 expression at baseline (Appendix A). Therefore, the absence of TRPV4 prevented the overexpression of TRPC6 during the promotion of pathological remodeling. Importantly, the response of TRPC6 to the activation of the CN/NFATc3 was intact in both TRPV4+/+ and TRPV4−/− mice: the activation of CN by extracellular Ca^2+^ overload translated into nuclear translocation of NFATc3 and TRPC6 overexpression in cardiac FB from animals of both genotypes. Conversely, CN inhibition with cyclosporine A (CsA, 1 µM) prevented NFATc3 translocation and TRPC6 overexpression (Figure 6E).

These results suggest that in the setting of pathological remodeling, TRPV4 channels are essential for the overexpression of TRPC6 channels, which can still be activated by the CN/NFATc3 pathway. To test the mechanism by which TRPV4 regulates TRPC6 expression, we first examined the ability of both channels to interact to form a functional ion channel. Our results confirmed the presence of TRPV4/TRPC1 complexes, as already reported [16], but did not detect the formation of heteromeric TRPV4/TRPC6 channels (Figure 6F). In this scenario, TRPV4 channels could be the primary elements promoting TRPC6 overexpression in pathological remodeling by activating the CN/NFATc3 pathway. To test this hypothesis, cultured FB of WT animals were exposed to the TRPV4 activator GSK1016790A in the absence or the presence of CsA. Interestingly, TRPC6 expression increased after exposure to the specific TRPV4 activator GSK1016790A (100 nM), but failed to do so when the CN/NFATc3 pathway was previously inhibited by CsA (1 µM) treatment (Figure 6G, left). TRPV4 expression, in contrast, was unchanged after GSK1016790A (100 nM) or GSK1016790A (100 nM) + CsA (1 µM) treatment (Figure 6G, right). Therefore, TRPV4 activation resulted in CN/NFAT-dependent upregulation of TRPC6 but had no effects on TRPV4 expression.

A new set of experiments were performed to better characterize the potential interplay between TRPV4 and TRPC6 and their role in the activation of the CN/NFAT pathway. Again, NFAT nuclear translocation was used as a surrogate of the activation of the pathway. Cultured FB from WT animals were exposed to the TRPV4 activator GSK1016790A (GSK10, 100 nM), the specific TRPC6 activator GSK1702934A (GSK17, 1 µM), or both. Additional conditions using specific inhibitors of TRPV4 (HC067047, HC, 10 µM) and TRPC6 (BI-749327, BI, 1 µM) were also assessed. The results of all experimental conditions are depicted in Figure 6H. Specific TRPV4 activation with GSK10 (purple bar) induced similar NFAT nuclear translocation to simultaneous activation of TRPV4 and TRPC6 (striped purple and yellow bar), which was greater than exclusive activation of TRPC6 (yellow bar). Importantly, inhibition of TRPC6 significantly attenuated the response to TRPV4 activation (striped purple and gray bar). Conversely, inhibition of TRPV4 while TRPC6 was activated (striped yellow and gray bar) induced a similar response to TRPC6 activation alone.

Altogether, our results indicate that as for TRPV4, TRPC6 overexpression is a specific finding of the pathological, but not the physiological, response and exhibits parallel behavior to that of TRPV4 during the development of maladaptive remodeling. Importantly, TRPV4 activation during pathological remodeling is necessary for TRPC6 overexpression to occur. In experimental conditions, TRPV4 channels can promote TRPC6 overexpression through activation of the CN/NFAT pathway, and the effects of TRPV4 activation on this pathway are enhanced in the presence of active TRPC6. Conversely, NFAT translocation induced by TRPC6 is independent of TRPV4 activation. These findings suggest that both TRPV4 and TRPC6 channels may participate synergistically in the pathological response, although only TRPV4 channels seem primary and essential elements for the combined effect of both channels (Graphical Abstract).

## 3. Discussion

This study provides several findings of novelty and special relevance: (1) TRPV4 channels mediate the development of pathological, but not physiological, remodeling, introducing a new differential element between both types of response; (2) during the development of HF, TRPV4 upregulation is associated with Ca^2+^-dependent activation of the CN/NFATc3 pathway in cardiac FB and secondary fibrosis; (3) TRPV4 activation during HF induces the upregulation of TRPC6, initiating a circuit of positive feedback that further enhances the maladaptive response in cardiac FB.

Most conditions causing chronic cardiac overload lead to the development of pathological or adverse remodeling following a period of compensatory hypertrophy. This response is mediated by G protein-coupled sympathetic and neurohormonal receptors, whose activation potentiates inotropism and cardiac growth in early phases but leads to the disruption of normal Ca^2+^ homeostasis and FB’s differentiation over time, initiating the transition to maladaptive remodeling [17]. Intracellular Ca^2+^ accumulation plays an important role in this process by activating Ca^2+^-dependent signaling pathways like CN/NFATc3 [18], calpain/IĸB/NF-ĸB [19], or CaMKII [20], among others. Conversely, cardiac remodeling may remain persistently balanced following certain conditions of cardiac overload, like chronic exercise. Several studies have pointed to the activation of the insulin-like growth factor 1 (IGF1)/PI3K/Akt/GSK3b cascade as a pivotal element in this type of response, known as adaptative or physiological [21]. The factors that trigger the preferential activation of one response or the other are not understood at present [22], although some authors have suggested that both the nature and the intensity of the primary stimulus generating cardiac overload are decisive [23].

Our study demonstrates that upregulation of TRPV4 (and secondarily, TRPC6) is an exclusive finding of pathological remodeling, introducing for the first time the potential role of mechanotransduction in determining the type of response. Importantly, our study further uncovered other differential traits between physiological and pathological remodeling. TRPV2 was overexpressed in the HF(iso) model, with no significant changes in the Ex model. A recent study showed that the deletion of TRPV2 has a protective role in pressure-overloaded hearts. However, this beneficial effect does not extend to AT-II or beta-adrenergic-induced cardiac remodeling [13], emphasizing the channel’s specificity in responding to particular mechanical conditions. Moreover, our findings revealed a consistent downregulation of TRPM4 and TRPV1 in response to exercise and a parallel (but non-significant) trend in beta-adrenergic-stimulated hearts, suggesting a potential regulatory role of these channels in both types of responses. Importantly, mice receiving chronic isoproterenol showed significant TRPV4 upregulation already at early stages (14 days of infusion), when hypertrophy was present, but ventricular dilation was only mildly increased and cardiac function was mildly impaired. In contrast, TRPV4 expression remained unchanged in mice subjected to exercise training despite showing a similar degree of chamber dilation. Therefore, our results would suggest that the factors triggering TRPV4 upregulation in pathological remodeling are not solely related to cell stretch but are more complex and probably involve chemical signals conditioned by the type of the stimulus.

During the development of pathological remodeling, TRPV4 overexpression increased mainly in cardiac FB and showed significant time-course correlation with collagen deposition, but did not precede fibrosis formation. This indicates that initial fibrosis, if mediated by TRPV4, should respond to increased activation of the channel at early stages rather than changes in expression. Alternatively, changes in extracellular matrix stiffness caused by initial collagen deposition could be the primary stimulus activating TRPV4 channels, which then would accelerate the progression of adverse remodeling by promoting further fibrosis. Although the exact primary element (TRPV4 activation or initial collagen deposition mediated by other factors) cannot be established from our work, in both scenarios, TRPV4 channels would accentuate the development of fibrosis, promoting adverse remodeling and ultimately HF.

The determining role of TRPV4 in fibrosis and the promotion of maladaptive remodeling was further confirmed by our results on transgenic mice. TRPV4−/− mice receiving isoproterenol treatment had preserved cardiac dimensions and function and exhibited significantly less collagen deposition and arrhythmia inducibility compared to TRPV4+/+. Therefore, under conditions of chronic β-adrenergic stimulation, the deletion of TRPV4 attenuated the main hallmarks of pathological remodeling. Recent data coming from experimental models and patients with dilated cardiomyopathy have confirmed that TRPV4 channels contribute to impaired cardiac contractility and HF progression by inducing abnormal cytosolic Ca^2+^ dynamics in CM [24,25]. The potential role of TRPV4 channels in cardiac FB was first suggested by Adapala et al., who demonstrated that TRPV4 is required for FB to differentiate into myofibroblasts in response to TGF-β [26]. The same group recently found that inhibition of TRPV4 protects against fibrosis formation following myocardial infarction by reducing FB’s differentiation through the mechanosensitive transcriptional Rho/Rho kinase/MRTF-A pathway [9]. Our study further confirms the pivotal role of TRPV4 in fibrosis formation in a model of chronic isoproterenol infusion aimed to better mimic the chronic adrenergic stimulation found in human HF [27] and provides preliminary data of a potential role of TRPV4 in conditions of pressure overload such as that induced by TAC. Moreover, we found that TRPV4 mediates fibrosis through activation of a signaling pathway different from that reported previously, the CN/NFATc3 pathway, a finding that further emphasizes the decisive role of TRPV4 in the development of cardiac fibrosis.

As cation channels have an affinity for Ca^2+^, it is fascinating to assess the role of TRPV4 in Ca^2+^-activated pathways, which have long-recognized implications in the promotion of adverse remodeling [5,17,18,19,20]. One such pathway is the CN/NFATc3 pathway. CN is a Ca^2+^-calmodulin-dependent serine/threonine phosphatase present in both CM and cardiac FB that, upon activation, causes NFATc3 dephosphorylation and translocation into the nucleus to promote, by interaction with the transcription factor GATA4, the expression of hypertrophic and fibrotic genes [14,28]. Transgenic mice that overexpress either active CN or a constitutively nuclear NFAT mutant protein develop cardiac hypertrophy and fibrosis, leading to HF [20]. Conversely, the use of CN and NFAT inhibitors in diverse experimental models of HF has shown to attenuate the maladaptive response [14,29,30]. Notably, among TRP channels, various TRPCs have been involved in the development of pathological remodeling through activation of the CN/NFATc3 pathway in CM [8,31]. Although TRPV4 channels are known to activate the same pathway in other cell types [15], to the best of our knowledge, ours is the first study to demonstrate their role in the activation of the Ca^2+^/CN/NFATc3 pathway in cardiac FB as a mechanism to explain the development of maladaptive remodeling.

One very relevant finding of our study is the potential regulatory role of cardiac TRPV4 channels as opposed to TRPC6. Several TRP channels are known to display functional interaction with members of the same or a different subfamily, or even form heterotetrameric complexes with unique functional properties. This is the case of co-assembly between TRPV4 and TRPV1/V2/V3, TRPC1, or TRPP2 subunits [32,33,34]. With regard to TRPV4 and TRPC6, a previous work found that both channels were frequently co-expressed in dorsal root ganglion neurons, potentially interacting in the hyperalgesic response [35]. However, we did not find heterotetramerization between both channels. In our study, expression of TRPV4 and TRPC6 channels, predominant in cardiac FB in either case, showed parallel behavior throughout the development of pathological remodeling and also in the differential pattern found in pathological versus physiological remodeling. Importantly, the activation of TRPC6 is known to induce CN/NFAT signaling and FB’s differentiation [36], which suggests that TRPV4 and TRPC6 could co-operate in the activation of this pathway to promote the fibrotic response under the appropriate overloading stimulus.

However, the role of TRPV4 would seem essential in this interaction. Transgenic TRPV4−/− mice did not show increased TRPC6 expression during HF, a response that was observed in TRPV4+/+ animals. NFAT is a known regulatory transcription factor of the trpc6 gene, at least in CM, [8] so we first confirmed that the absence of TRPC6 was not constitutional in the transgenic colony by activating the CN/NFAT pathway with Ca^2+^ overload in cultured FB, which did induce TRPC6 overexpression. Interestingly, TRPC6 was also upregulated in cardiac FB after exposure to the selective TRPV4 activator GSK1016790A, a response that was blunted when CN was inhibited by CsA. These results suggest that during the development of pathological remodeling, TRPV4 channels are necessary for TRPC6 overexpression to occur, probably by activating the CN/NFATc3 pathway and consequently upregulating TRPC6, which in turn would further enhance the same pathway (Graphical Abstract). As such, NFAT translocation is maximal when TRPV4 channels are activated (and similar to that observed after simultaneous activation of TRPV4 and TRPC6), but is significantly reduced when TRPV4 channels are activated in the presence of TRPC6 inhibitors. Importantly, this seems to be a unidirectional effect, since NFAT activation by TRPC6 is not modified by the activation status of TRPV4. Altogether, these findings would pose TRPV4 channels as the primary mechanoreceptors initiating the pathological response.

Our study has several limitations. We did not study the effects of TRPV4 deletion on physiological remodeling. Given the lack of TRPV4 upregulation in both models of exercise, and invoking the ethical principles of the 3Rs on animal research, transgenic mice were exclusively used for the study of pathological remodeling. Furthermore, by using a transgenic model, we only assessed the effects of TRPV4 inhibition on preventing the development of pathological remodeling. Whether TRPV4 can also have a role in reverting the HF phenotype when already present remains to be determined. We focused on the study of TRPV4 deletion in cardiac FB given that this work was conceived to assess the mechanisms of mechanotransduction in cardiac fibrosis, and on the other hand, given that these channels were majorly found in this cell type. However, since our work is based on global TRPV4 knockout mice, the broader implications of TRPV4 in other cell types, such as in CM or in vascular or inflammatory cells, merit further investigation. Previous studies suggest that TRPV4 could also have an effect on CM [24,25], and our results indicate that TRPV4 inhibition reduces CM hypertrophy, either by direct effect on these cells or by modulatory cross-talking between FB and CM. Additionally, recent research proposes that deleting TRPV4 in endothelial cells might preserve cardiac function by enhancing coronary angiogenesis [37]. We did not assess the mechanisms by which arrhythmia inducibility was reduced in TRPV4−/− mice compared to TRPV4+/+ mice, which could be explained by mere reduction in the arrhythmogenic substrate (i.e., fibrosis) or by specific antiarrhythmic mechanisms as opposed to CM, as suggested by recent reports [38]. Antibody and isoform specificity were validated using the KO model for TRPV4. However, due to the unavailability of a TRPC6 KO control, direct assessment of isoform specificity for TRPC6 was not feasible. To address this limitation, the samples were incubated with a blocking peptide. Finally, further research is needed to ascertain whether other mechanisms are involved in the interaction between TRPV4 and TRPC6 regarding fibrosis and HF promotion.

In conclusion, our study shows that TRPV4 channels mediate exclusively the development of pathological remodeling, promoting tissue fibrosis through Ca^2+^-dependent activation of the CN/NFATc3 pathway and enhancing TRPC6 upregulation. These results suggest that TRPV4 could be a potential primary mediator of the pathological response, pointing to a valuable target for the treatment of HF.

## 4. Materials and Methods

An extended version of the Materials and Methods section is available in the Appendix A.

### 4.1. Animals

Experiments were performed in 10-week-old C57BL/6J mice (25–30 g) and 8-week-old Wistar rats (200–250 g). Additionally, transgenic C57BL/6J wildtype (WT, TRPV4+/+) and TRPV4 knockout (KO, TRPV4−/−) mice at the age of 10 weeks were used in some experiments. The TRPV4 transgenic colony was generated by Cre-Lox-mediated excision of exon 12, which induces a constitutive deletion of the trpv4 gene, as previously reported [39].

### 4.2. In Vivo Models of Adverse and Adaptive Cardiac Remodeling

Adverse remodeling was induced in mice by continuous infusion of the β-adrenergic agonist (isoproterenol) through subcutaneous osmotic minipumps (1004 Alzet, Cupertino, CA, USA) set to release either the active drug (isoproterenol 30 mg/kg/day, HF(iso) model) or saline (0.9% NaCl, sham) over 28 days, as previously described [40]. Physiological remodeling was induced by subjecting mice to moderate exercise training (Ex) on a treadmill (15 cm/s, 6° positive slope) for 30 min, 5 days a week, for 8 weeks, using an adapted protocol from previous publications [41]. Sedentary animals (Sed), housed and fed in the same conditions, served as controls (Appendix A).

Maladaptive and adaptive remodeling were also assessed in two rat models that served as external controls. Maladaptive remodeling was generated by transverse aortic constriction (TAC), achieved by ligating a 5.0 nylon suture around the aortic arch, as previously described [42]. Sham animals underwent thoracic surgery with no ligation. Cardiac remodeling was assessed in all animals 8 weeks after surgery. Adaptive remodeling in rats was achieved by moderate exercise training on a treadmill (35 cm/s, 0° slope) for 45 min, 5 days a week, for 16 weeks. Sedentary animals remained with no training in the same conditions (Appendix A).

At their final endpoint, animals were euthanized by intraperitoneal (IP) injection of sodium pentobarbital (100 mg/kg). The hearts were removed and either cannulated on a Langendorff perfusion system, weighted and fixed for histology, or snap frozen in liquid nitrogen. All stored samples were kept at −80 °C.

### 4.3. Echocardiography Measurements

Serial echocardiograms (Vivid IQ system, General Electric Healthcare, Horten, Norway) were performed to measure the parameters of LV dimensions and function (LVD, IVS, and LVEF).

### 4.4. Histology

Heart samples fixed in buffered 4% formaldehyde were embedded in paraffin and cut into 4 µm thick slices. LV cardiomyocyte cross-sectional area (CM-CSA) was measured in transverse sections stained with hematoxylin and eosin (H&E), and LV fibrosis was assessed by quantifying collagen deposition in heart sections stained with picrosirius red, as previously described [41].

### 4.5. Langendorff-Perfused Hearts

Immediately after sacrifice, the hearts were cannulated via the aorta into a standard Langendorff perfusion system, secured with a 3-0 suture, and kept beating with retrograde perfusion through the aorta with a constant flow of an oxygenated (95% O_2_; 5% CO_2_) Krebs solution at 37 °C (118 mM NaCl, 4.7 mM KCl, 1.2 mM MgSO_4_, 1.8 mM CaCl_2_, 25 mM NaHCO_3_, 1.2 mM KH_2_PO_4_, and 11 mM glucose, pH 7.4). The setup was adjusted to produce a perfusion pressure of 80–90 mmHg (normoxic environment). Left ventricular pressure was monitored using a water-filled latex balloon placed in the LV, connected to a pressure transducer.

### 4.6. Assessment of Arrhythmias

Arrhythmia inducibility was assessed in Langendorff-perfused hearts under normoxic conditions and also after inducing regional ischemia through ligation of the left anterior descending artery, using a modified version of a previously validated stimulation protocol [43,44].

### 4.7. Fibroblast Culture

CM and FB were isolated from hearts hung in a Langendorff perfusion system through specific digestion (see Appendix A and Appendix A). Freshly isolated FB were cultured in DMEM medium (#30-2002; ATCC, Manassas, VA, USA) supplemented with 10% FBS and 2% penicillin/streptomycin for 2 h at 37 °C and 5% CO_2_, and washed afterwards. Only viable FB, attached to the plate, were kept, while other cells were washed out. The medium was changed after 24 h, and fibroblasts were allowed to grow until confluence.

### 4.8. qPCR Analyses

qPCR analyses were performed with LV samples, freshly isolated FB, or cultured FB by extracting RNA with the Nucleospin RNA extraction kit (Macherey-Nagel, Düren, Germany), which was then retrotranscribed into cDNA with a High-Capacity cDNA Reverse Transcription Kit (Applied Biosystems, Waltham, MA, USA). In a set of experiments aimed to study the effects of TRPV4 activation on TRPV4 and TRPC6 expression, FB from WT animals were treated for 4 h with GSK1016790A (GSK, 100 nM), either in the absence or presence of the calcineurin inhibitor cyclosporine A (CsA, 1 µM, 1 h before GSK challenge), before total RNA was extracted. Gene expression was measured in triplicate with a 7900HT Fast Real-Time PCR System (Applied Biosystems, Waltham, MA, USA) using TaqMan Universal PCR master mix (Thermofisher, Waltham, MA, USA) and pre-designed gene-specific probes (Thermofisher, Waltham, MA, USA; see Appendix A). Relative quantification was calculated using the comparative threshold method and expressed as fold change (FC) over the control group.

### 4.9. Western Blotting

Total protein from tissues and cells was extracted with RIPA buffer (50 mM Tris Base, 150 mM NaCl, 10 mM EDTA, 0.1% SDS, 0.5% Na-deocycholate, 1% Triton-X-100, 10 mM NaF, 2 mM Na_3_VO_4_, and 1% protease inhibitor, pH 7.3). Protein samples were separated in 10% SDS-PAGE gel electrophoresis, transferred to nitrocellulose membranes, and blocked with 5% non-fat milk TTBS. Membranes were probed overnight at 4 °C with the following antibodies, TRPV4 (ACC034, 1:500; Alomone labs, Jerusalem, Israel), TRPC6 (PA5-77308, 1:500, or PA5-29848, 1:1000; Thermofisher, Waltham, MA, USA), CaN (610259, 1:500; BD Biosciences, Franklin Lakes, NJ, USA), and GAPDH (GT239, 1:10,000; GeneTex, Irvine, CA, USA), as the endogenous control. Antibody and isoform specificity were validated using the KO model for TRPV4. However, due to the unavailability of a TRPC6 KO control, direct assessment of isoform specificity for TRPC6 was not feasible. To address this limitation, the samples were incubated with a blocking peptide. After three washes, membranes were incubated with peroxidase-conjugated secondary anti-rabbit or mouse IgG for 1 h at room temperature. After washing, the proteins were developed with ECL (Amersham Biosciences, Amersham, UK) and captured using an Odyssey FC Imaging System (LI·COR Biosciences, Licoln, NE, USA). Band intensities were measured by densitometry scanning with Image Studio Lite software (version 5.2).

### 4.10. Fluorescence Ca^2+^ Imaging

FB were loaded with Fluo-4/AM (5 µM) for 30 min at 37 °C, washed, and exposed to an isotonic solution (~300 mOsm). Basal intracellular Ca^2+^ (F0) was monitored every 15 s for 90 s with a microplate reader SpectraMax ID3 (Molecular Devices, San Jose, CA, USA). Cells were then incubated with the TRPV4 pharmacological activator GSK1016790A (100 nM) or hypotonicity (~140 mOsm solution), a physiological activator, and Ca^2+^ measurements were continued every 3 s up to 3 min. In some experiments, cells were pre-treated with the TRPV4 antagonist HC067047 (HC; 10 µM) for 3 min. Changes in intracellular Ca^2+^ were calculated as the ratio of Fluo-4/AM fluorescence intensity at each timepoint relative to the basal fluorescence (F/F0).

### 4.11. Calcineurin Activity Assay

The enzymatic activity of calcineurin was measured in protein extracts collected from isolated FB using the calcineurin cellular activity assay kit (BML-AK816-0001; Enzo Life Sciences, Farmingdale, NY, USA). Calcineurin activity was quantified by measuring the absorbance of malachite green (OD 630 nm) of the released free phosphates.

### 4.12. NFAT Nuclear Translocation

Cytosolic/nuclear localization of NFAT was assessed by immunofluorescence, and NFAT translocation was defined as the ratio of mean fluorescence intensity at the nucleus relative to the cytosol in different experimental conditions (see Appendix A). At least 30 isolated FB were measured per condition in 4 independent experiments.

### 4.13. Proximity Ligation Assay

Proximity ligation assay (PLA) was used to examine the formation of heteromeric channels. Cultured isolated CFb were fixed, permeabilized, and incubated with rabbit anti-TRPV4 (ab191580, Abcam, Cambridge, UK), rabbit anti-TRPC1 (PA5-100358, Thermofisher, Waltham, MA, USA), and rabbit anti-TRPC6 (PA5-29848, Thermofisher, Waltham, MA, USA), previously conjugated to PLA probes using the Duolink in situ Probemaker PLUS (DUO92010, Thermofisher, Waltham, MA, USA) or MINUS (DUO92009, Thermofisher, Waltham, MA, USA) kit. Detection was performed with Duolink in situ detection reagent red kit (DUO92008, Thermofisher, Waltham, MA, USA) following the manufacturer’s protocol, and cells were imaged using a Zeiss LSM980 with a 60X objective.

### 4.14. Statistical Analysis

Data are presented as mean ± standard error of the mean (SEM). Statistical analyses were performed using *t*-test or one-way ANOVA for variables respecting a Gaussian distribution. Two-way ANOVA was used in most of the experiments performed in transgenic mice to evaluate the effects of TRPV4 deletion, where factors were genotypes (TRPV4+/+ and TRPV4−/−) and treatment group (sham and iso). Since arrhythmia inducibility did not follow a Gaussian distribution, differences during normoxia and after ischemia were assessed by a non-parametric equivalent of a two-way ANOVA test (Scheiner–Ray–Hare, SRH test). All ANOVA and non-parametric analyses were followed by a Bonferroni post hoc correction when interaction was found. Data were analyzed using GraphPad Prism 6.0 and differences were considered statistically significant when *p* < 0.05.

## Figures and Tables

**Figure 1 ijms-25-01541-f001:**
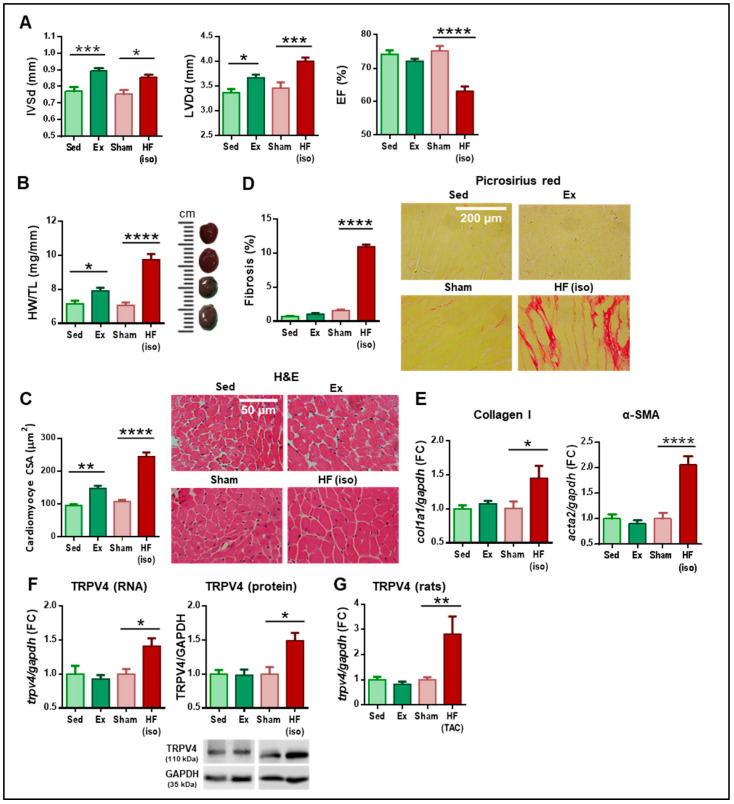
Differential features between physiological (green) and pathological (red) remodeling. From (**A**–**G**), results of the four experimental groups generated in mice (N = 6–12/group). (**A**) Echocardiographic parameters of interventricular septum thickness in diastole (IVSd), left ventricular diameter in diastole (LVDd), and ejection fraction (EF). (**B**) Heart weight-to-tibial length (HW/TL) ratio. On the right, representative images visually depict the size of the heart. (**C**) Overall quantification of cardiomyocyte cross-sectional area (CSA) with representative photomicrographs of the four study groups stained with hematoxylin and eosin (H&E). Scale bar represents 50 µm. (**D**) Percentage of fibrosis measured by collagen deposition with representative images of the four study groups stained with picrosirius red. Scale bar corresponds to 200 µm. (**E**) Gene expression of fibrotic markers in the four study groups. (**F**) Gene and protein expression of TRPV4 in the four study groups. (**G**) Gene expression of TRPV4 in the four experimental groups studied in rats (see the Methods section, N = 12/group). Ex: exercise; Sed: sedentary; HF(iso): HF induced by isoproterenol infusion; HF(TAC): HF induced by transverse aortic constriction. * *p* < 0.05; ** *p* < 0.01; *** *p* < 0.001; and **** *p* < 0.0001.

**Figure 2 ijms-25-01541-f002:**
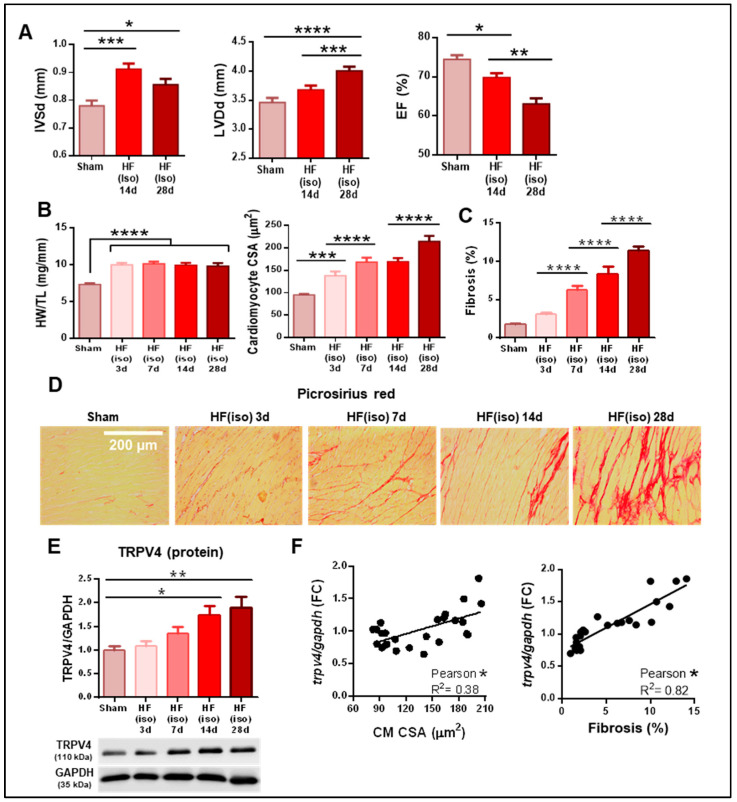
Time-course changes during the development of pathological remodeling induced by isoproterenol infusion (N = 5–18/group). (**A**) Echocardiographic parameters of interventricular septum thickness in diastole (IVSd), left ventricular diameter in diastole (LVDd), and ejection fraction (EF). (**B**) Heart weight-to-tibial length ratio (HW/TL, left) and cardiomyocyte cross-sectional area (CSA, right). (**C**) Fibrosis quantification by percentage of collagen deposition. (**D**) Representative microphotographs stained with picrosirius red at each timepoint. Scale bar corresponds to 200 µm. (**E**) Protein expression of TRPV4 channels over time. (**F**) Correlation of TRPV4 expression with cardiomyocyte CSA (left) and collagen deposition (right) at all timepoints. Iso 3d, 7d, 14d, and 28d represent mice subjected to isoproterenol infusion for 3, 7, 14 and 28 days, respectively. * *p* < 0.05; ** *p* < 0.01; *** *p* < 0.001; and **** *p* < 0.0001.

**Figure 3 ijms-25-01541-f003:**
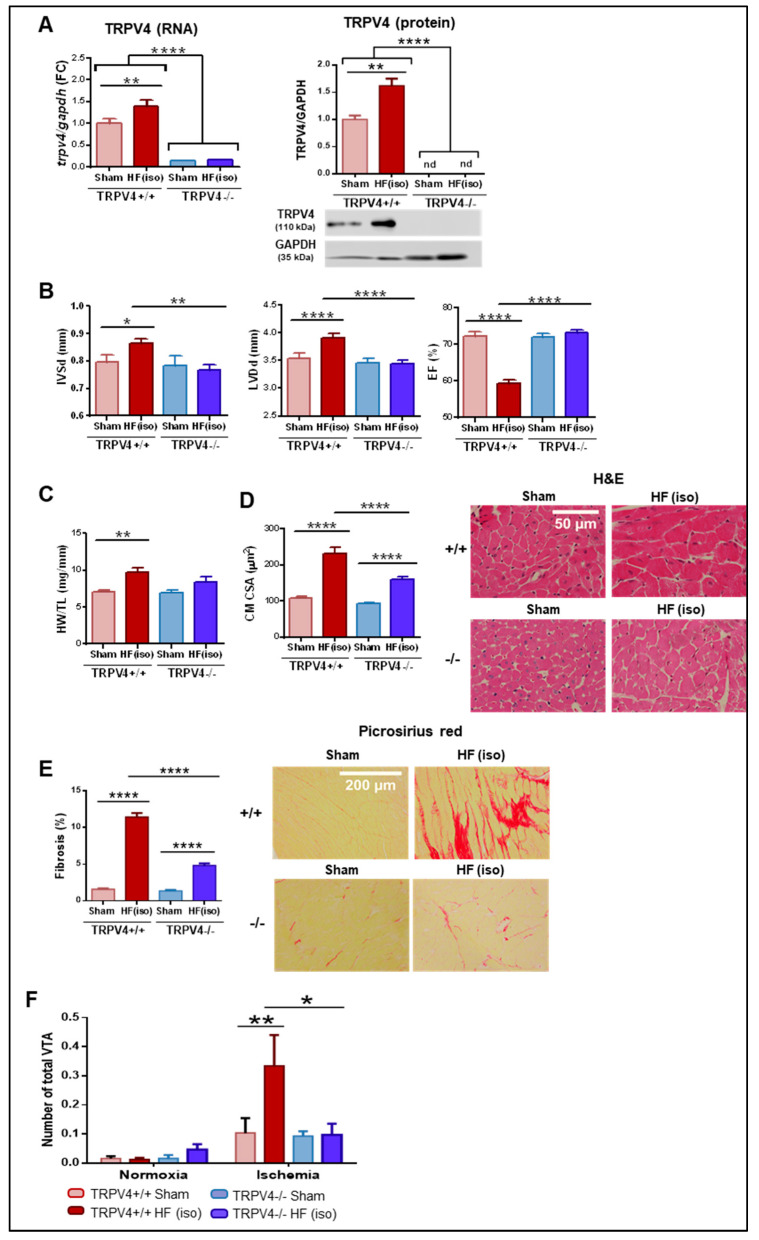
Induction of pathological remodeling in TRPV4+/+ and TRPV4−/− mice (N = 6–12/group). (**A**) Gene and protein expression of TRPV4 channels in the four study groups. (**B**) Echocardiographic parameters of interventricular septum thickness in diastole (IVSd), left ventricular diameter in diastole (LVDd), and ejection fraction (EF). (**C**) Heart weight-to-tibial length (HW/TL) ratio. (**D**) Overall quantification of cardiomyocyte cross-sectional area (CSA), with representative hematoxylin and eosin (H&E) stained images of all experimental groups. Scale bar represents 50 µm. (**E**) Fibrosis quantification by percentage of collagen deposition with representative microphotographs stained with picrosirius red. Scale bar corresponds to 200 µm. (**F**) Arrhythmia inducibility under normoxia and ischemia. The number of total ventricular tachyarrhythmias (VTA) are shown for each condition. HF(iso): HF induced by isoproterenol infusion; nd: not detected; * *p* < 0.05; ** *p* < 0.01; and **** *p* < 0.0001.

**Figure 4 ijms-25-01541-f004:**
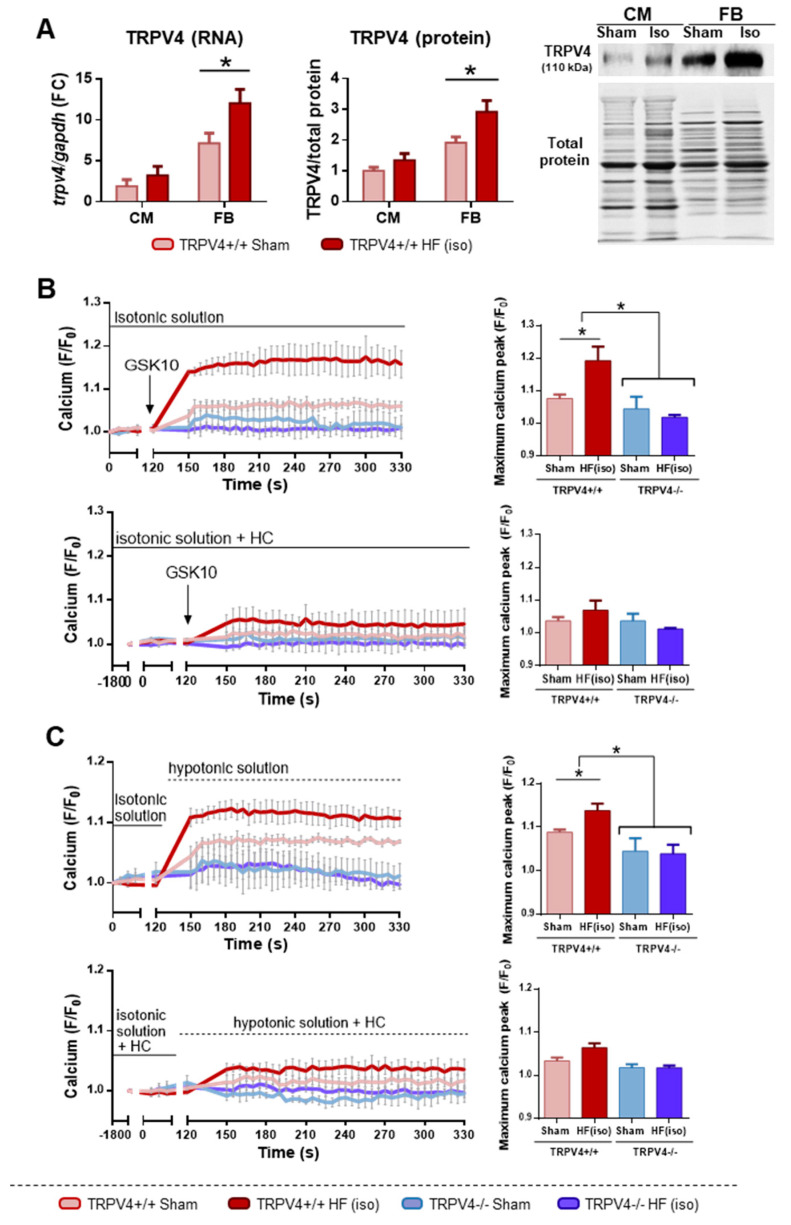
Functional dynamics of Ca^2+^ in response to specific TRPV4 activation and/or inhibition (N = 4–8 replicates per group from 3–4 independent experiments). (**A**) Relative expression of TRPV4 in cardiomyocytes (CM) and fibroblasts (FB). (**B**) Calcium influx recorded in FB from TRPV4+/+ and TRPV4−/− mice (groups HF(iso) and sham) in response to the selective TRPV4 activator GSK1016790A (GSK10, 100 nM) in the absence (left) or the presence (right) of the TRPV4 inhibitor HC067047 (HC, 10 µM). (**C**) Calcium influx in response to a hypotonic solution (140 mOsm) in the absence (left) or the presence (right) of the TRPV4 inhibitor HC067047 (HC, 10 µM). F/F0 = ratio of fluorescence intensity relative to time 0; HF(iso): HF induced by isoproterenol infusion. In (**B**,**C**), all Ca^2+^ measures are expressed as the change with respect to baseline values, which have been normalized to 1. * *p* < 0.05.

**Figure 5 ijms-25-01541-f005:**
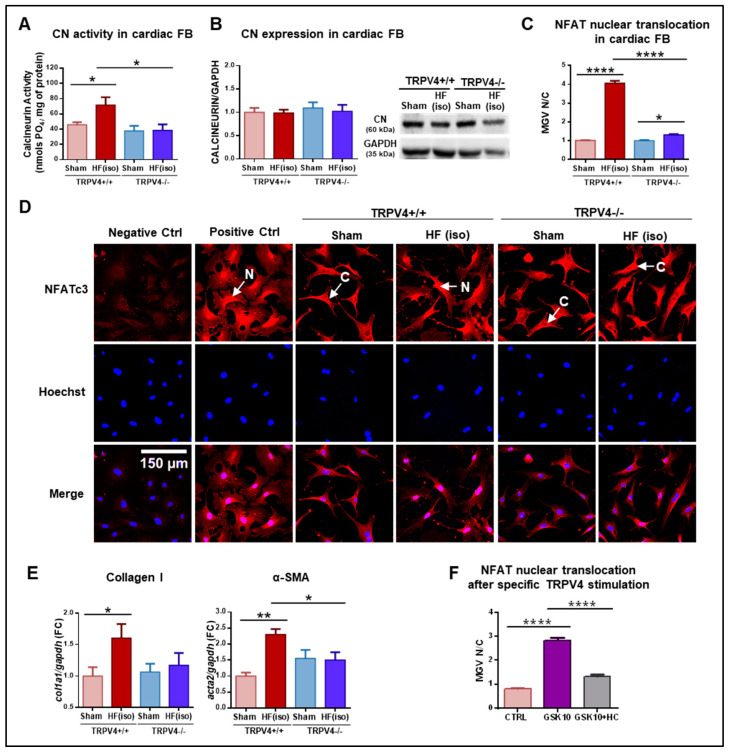
(**A**) Calcineurin (CN) activity and (**B**) protein expression in isolated FB from TRPV4+/+ and TRPV4−/− mice of the HF(iso) and sham groups (N = 4–5/group). (**C**) NFAT cytoplasmatic and nuclear fluorescence quantification expressed as the MGV N/C, the ratio of mean fluorescence intensity (mean gray value) between the nucleus (N) and the cytoplasm (C), n = 202–244. (**D**) Representative confocal microscopy images of all experimental groups. From top to bottom, each row represents NFAT staining (red), nuclear staining (blue), and merging. The negative control was not incubated with the primary antibody. The positive control was obtained by activation of CN following incubation with a high-calcium medium (4 mM). Scale bar corresponds to 150 µm. (**E**) Gene expression of col1a1 and acta2 in the four study groups. (**F**) NFAT nuclear translocation in isolated FB from TRPV4+/+ mice (WT), expressed as the MGV N/C, after specific TRPV4 stimulation with GSK101679A (GSK10 (100 nM), purple), and GSK + pre-incubation with the TRPV4 inhibitor HC067047 (HC (10 µM), gray) in isolated FB. * *p* < 0.05; ** *p* < 0.01; and **** *p* < 0.0001.

**Figure 6 ijms-25-01541-f006:**
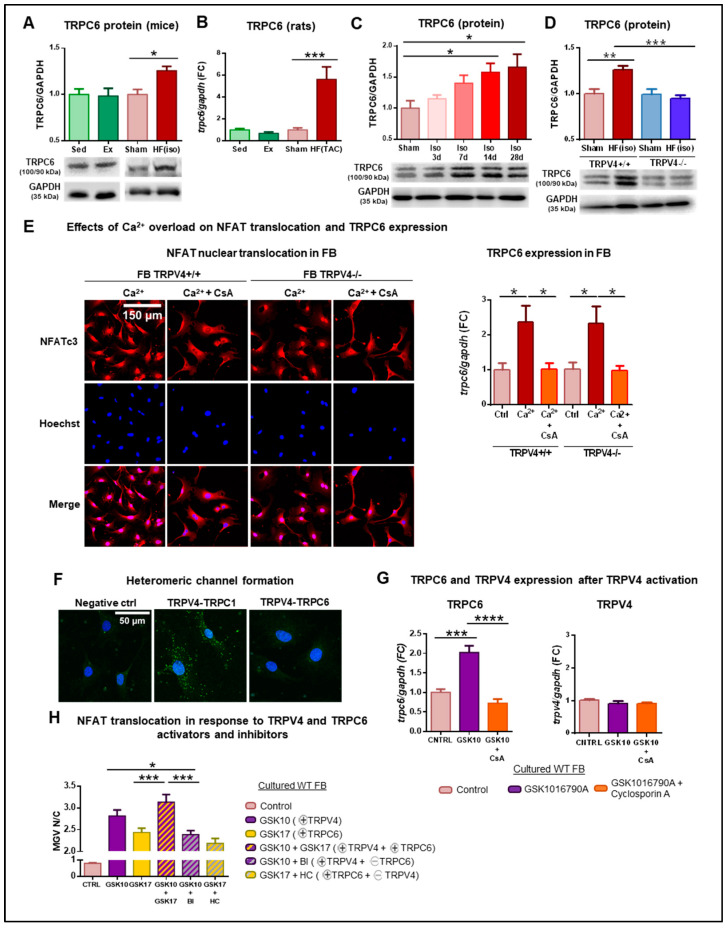
(**A**) TRPC6 protein expression in the HF and Ex models induced in mice (n = 5/group). (**B**) Gene expression of TRPC6 in the four experimental groups studied in rats (N = 12/group). (**C**) Time-course protein expression of TRPC6 channels during the development of pathological remodeling (N = 5–11/group). (**D**) TRPC6 protein expression in TRPV4+/+ and TRPV4−/− mice receiving isoproterenol (HF(iso)) or saline (sham) (N = 6–8/group). (**E**) Effects of Ca^2+^ overload and calcineurin inhibition with cyclosporine A (CsA) on NFATc3 nuclear translocation (left) and TRPC6 gene expression (right) in FB from TRPV4+/+ and TRPV4−/− mice from the HF(iso) and sham groups. (**F**) Proximity ligation assay (PLA) for heteromeric channel formation of TRPV4-TRPC1 and TRPV4-TRPC6. PLA was performed with a combination of anti-TRPV4, anti-TRPC1, and anti-TRPC6 antibodies conjugated to PLA PLUS or MINUS probes. The negative control was only incubated with PLA PLUS and MINUS probes. (**G**) In TRPV4+/+ mice (WT), TRPC6 (right) and TRPV4 (left) expression following exposure to the specific TRPV4 activator GSK10 (100 nM) without and with pre-incubation with CsA (1 µM), a CN inhibitor (N = 5–6/group). (**H**) NFAT nuclear translocation as a surrogate of activation of the CN/NFAT pathway in cardiac FB from TRPV4+/+ (WT) animals; experimental conditions were as follows: TRPV4 activation by GSK1016790A (GSK10, 100 nM), TRPC6 activation by GSK1702934A (GSK17, 1 µM), simultaneous activation of TRPV4 and TRPC6 (GSK10, 100 nM + GSK17, 1 µM), TRPV4 activation with previous inhibition of TRPC6 (GSK10, 100 nM + BI-749327, 1 µM), TRPC6 activation with previous inhibition of TRPV4 (GSK17, 1 µM + HC067047, 10 µM), and control. All conditions were significantly different (*p* < 0.0001) compared to controls. Ex: exercise; Sed: sedentary; HF(iso): HF induced by isoproterenol infusion; HF(TAC): HF induced by transaortic constriction; * *p* < 0.05; ** *p* < 0.01; *** *p* < 0.001; and **** *p* < 0.0001.

## Data Availability

Data is contained within the article and Appendix A.

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
