# Peer review of "TRPV4 Channels Promote Pathological, but Not Physiological, Cardiac Remodeling through the Activation of Calcineurin/NFAT and TRPC6"

_ijms, 2024, doi:10.3390/ijms25031541_

Round 1

Reviewer 1 Report

Comments and Suggestions for Authors

General comment:

This report by Yáñez-Bisbe et al. provides evidence supporting the key role of TRPV4 in cardiac hypertrophy and heart failure. The authors add a new aspect to this concept. Their data suggest a specific, pathophysiologic function of TRPV4 in cardiac fibroblasts and hypertrophy-associated cardiac fibrosis. It is demonstrated that maladaptive but not physiological remodeling leads to enhanced cardiac TRPV4 expression, with a time course that correlates well with fibrosis development and fibrosis-associated dysfunction. The authors use established, murine models of heart failure development along with genetic ablation of TRPV4 and complement this approach with experiments in isolated cardiac fibroblasts. The authors conclude that TRPV4 is prominently (primarily) upregulated in cardiac fibroblasts, thereby enhancing Ca2+ entry that triggers NFATc3 activation and consequently enhanced expression of TRPC6. The authors propose a synergistic, pathophysiologic function of TRPV4 and TRPC6 in cardiac fibroblasts. The report is certainly interesting and timely, nonetheless, I have a number of technical and conceptual problems and concerns as listed below.

Specific comments and suggestions:

1)    The authors test the hypothesis of a critical role of TRPV4 in cardiac fibroblasts by characterizing fibroblast cultures derived from cardiac hypertrophy / HF models. Overall, the results indicate that there may be enhanced TRPV4-mediated Ca2+ entry into the fibroblasts leading to increased calcineurin activity and NFATc3 translocation. I have several problems with these conclusions and the underlying mechanistic concept. The authors demonstrate TRPV4-mediated Ca2+ signals in the fibroblast (Figure 4), which are, however, triggered by chemical and osmotic stimuli. On the other hand, the authors report apparently constitutively active NFATc3 in their fibroblast cultures from HF hearts (Figure 5). Similarly, calcineurin (CN) activity is found enhanced, constitutively, in fibroblast cultures. This finding is quite puzzling in view of the known transient nature of especially NFATc3 translocation. One explanation would be a high constitutive activity of TRPV4 in these cultures even in the absence of stress factors. This needs to be further tested. Is the in vitro observed CaN and NFATc3 activity indeed sensitive to TRPV4 inhibition? Is NFATc3 activation also observed at the level of dephosphorylation? 

2)    The authors compare isolated cardiomyocytes (CM) and cultured fibroblasts in terms of function (CaN activity). Is this a reasonable comparison. What is the quality of the CM preparation? Is this a pure and viable CM preparation? Please provide a documentation to support this set of results.

3)    It is well established that TRPV4 is significantly expressed in vascular endothelium and myocytes, and a pathophysiological role of vascular TRPV4 has been proposed. Can the authors exclude a role of vascular TRPV4 and TRPC6 in their observations? The potential role of TRPV4 in vascular cells need a more comprehensive and reasonable discussion. 

4)    The authors emphasize that their results demonstrate an exclusively pathological role of TRPV4 in cardiac remodeling. It appears that this differential role (physiological vs pathophysiological) in cardiac remodeling holds true also for other TRP channels. Please discuss.

5)    A critical tool in this study are commercially available antibodies, which vary considerably from batch to batch in terms of quality. How did the authors test and confirm antibody specificity? This needs some documentation in the methods section, specifically for the TRPC6 antibody.

Minor: 

1)    please state drug (inhibitor and activator) concentrations in figure legends

2)    the manuscripts needs a careful check for typos          

Reviewer 2 Report

Comments and Suggestions for Authors

The manuscript by Yannez-Bisbe et al. described the involvement of TRPV4 channel in pathological cardiac remodeling though Ca2+-depending signalling involving calcineurin/NFATc3 pathway activation.

I found this study very interesting. Authors have used numerous different approaches (echocardiography and histology measurements, Langendorff-perfused heart experiments, Ca2+ measurements, immunofluorescence, proximity ligation assays…) to validate their hypotheses. The experiments are carefully performed, their interpretations are accurate, and the paper is very well written.

I just have few comments that should be addressed in the final manuscript.

1/ In the Fluorescence Ca2+ imaging experiments (as mentioned in the supplemental material), the authors add thapsigargin to inhibit Ca2+ reuptake to the endoplasmic reticulum and to enhance changes in cytosolic Ca2+. This protocol is questionable: in addition to induce a basal capacitive Ca2+ influx, the kinetics of the calcium responses induced by the GSK and the hypotonic solution are modified. Would the recorded Ca2+ responses (in Figure 4) be sustained in the absence of thapsigargin? This should at least be discussed in the text.

2/ Lines 333-335: "activation of CN by extracellular Ca2+ overload translated into nuclear translocation of NFATc3 and TRPC6 overexpression in cardiac FB from animals of both genotypes (Figure 6E)". In my opinion the link between nuclear translocation of NFATc3 and TRPC6 overexpression is not established. The authors should complete this experiment by adding cyclosporin A.

3/ Why do TRPV4 expression quantification experiments carried out in rats only involve mRNA and not protein (Figures 1G and 6B), as was done in mice?

4/ The concentrations of all agonists, antagonists, and inhibitors (GSK1016790A, GSK1702934A, HC067047, BI-749327, cyclosporinA…) should appear in the text. Unless I am mistaken, only the concentrations of GSK1016790A and HC067047 appear in the "Materials and Methods" section, in the paragraph "Fluorescence Ca2+ imaging".

5/ In my opinion, all figures are too small. In addition, all figure legends need to be modified: the lower-case letters in the legends do not correspond to the upper-case letters in the panels.

6/ Figure 1B: the representative images in the left are not mentioned in the legend.

7/ There are a number of small mistakes throughout the text that need to be corrected:

- in many lines: "Ca2+" instead of "Ca2+"

- line 100: delete "18" between "heart" and "including"

- line 277: alpha-SMA

- line 311: delete "8" between "feedback" and "We"

- lines 348 and 349: "Figure 6G" instead of "Figure 6F" (same remark in the "Supplemental material", in the "qPCR analyses" section)

- line 359: "Figure 6H" instead of "Figure 6G"

- lines 376 and 506: delete "(see graphical abstract)" or add this graphical abstract

- line 473: delete "20" between "HF" and "Conversely"

- line 561: delete "15" between "weeks" and "Sedentary"

Round 2

Reviewer 1 Report

Comments and Suggestions for Authors

The authors have done a good job to improve their manuscript and increased significance of this study.

I have only one objection/critcism remaining. Previous mayor point 5: It is indeed good that there is a blocking peptide available, and that "general suitability" of the AB has been tested by use of the "blocking" peptide. However, this (standard control) approach does not provide any information regarding the actual TRP isoform specificity or even specificity vs other related sequences. In case the authors cannot confirm/test specificity by use of a suitable KO model, I suggest to clearly state the limitation in both the methods section and the discussion.

Reviewer 2 Report

Comments and Suggestions for Authors

Overall, I am satisfied with the authors' responses to my comments.

However, the manuscript needs to be carefully checked, as there are still errors that have not been corrected despite my previous comments:
- line 100: delete "18" between "heart" and "including",
- "Figure 6G" instead of "Figure 6F" in the "Supplemental material", in the "qPCR analyses" section,
- lines 378 and 516: delete "(see graphical abstract)" or add this graphical abstract;
and new ones have appeared:
- lines 336-352: "µM" or "nM" intead of "µm" or "nm"
- line 426: "Ca2+" instead of "Ca2+"

In addition, the figures have been slightly enlarged but their quality has been degraded.
